# A Smooth Binary Mechanism for Efficient Private Continual Observation

**Joel Daniel Andersson**
Basic Algorithms Research Copenhagen
University of Copenhagen
jda@di.ku.dk

**Rasmus Pagh**
Basic Algorithms Research Copenhagen
University of Copenhagen
pagh@di.ku.dk

## Abstract

In privacy under continual observation we study how to release differentially private estimates based on a dataset that evolves over time. The problem of releasing private prefix sums of $x_1, x_2, x_3, \cdots \in \{0, 1\}$ (where the value of each $x_i$ is to be private) is particularly well-studied, and a generalized form is used in state-of-the-art methods for private stochastic gradient descent (SGD). The seminal *binary mechanism* privately releases the first $t$ prefix sums with noise of variance polylogarithmic in $t$. Recently, Henzinger et al. and Denisov et al. showed that it is possible to improve on the binary mechanism in two ways: The variance of the noise can be reduced by a (large) constant factor, and also made more even across time steps. However, their algorithms for generating the noise distribution are not as efficient as one would like in terms of computation time and (in particular) space. We address the efficiency problem by presenting a simple alternative to the binary mechanism in which 1) generating the noise takes constant average time per value, 2) the variance is reduced by a factor about 4 compared to the binary mechanism, and 3) the noise distribution at each step is *identical*. Empirically, a simple Python implementation of our approach outperforms the running time of the approach of Henzinger et al., as well as an attempt to improve their algorithm using high-performance algorithms for multiplication with Toeplitz matrices.

## 1 Introduction

There are many actors in society that wish to publish aggregate statistics about individuals, be it for financial or social utility. Netflix might recommend movies based on other users' preferences, and policy might be driven by information on average incomes across groups. Whatever utility one has in mind however, it should be balanced against the potential release of sensitive information. While it may seem anodyne to publish aggregate statistics about users, doing it without consideration to privacy may expose sensitive information of individuals (Dinur & Nissim, 2003). Differential privacy offers a framework for dealing with these problems in a mathematically rigorous way.

A particular setting is when statistics are updated and released continually, for example a website releasing its number of visitors over time. Studying differential privacy in this setup is referred to as *differential privacy under continual observation* (Dwork et al., 2010; Dwork & Roth, 2013). A central problem in this domain is referred to as *differentially private counting under continual observation* (Chan et al., 2011; Dwork et al., 2010), *continual counting* for short. It covers the following problem: a binary stream $x_1, x_2, x_3, \ldots$ is received one element at a time such that $x_t$ is received in round $t$. The objective is to continually output the number of 1s encountered up to each time step while maintaining differential privacy. We consider two streams $x$ and $x'$ as neighboring if they are identical except for a single index $i$ where $x_i \neq x'_i$. It suffices to study the setting in which there is a known upper bound $T$ on the number of prefix sums to release — algorithms for the case of unbounded streams then follow by a general reduction (Chan et al., 2011).

37th Conference on Neural Information Processing Systems (NeurIPS 2023).

*Table 1: Comparison between different $\rho$-zCDP mechanisms for continual counting.*

| Mechanism $\mathcal{M}$ | $\mathrm{Var}[\mathcal{M}(t)] \cdot \frac{2\rho}{\log_2^2 T}$ | Time to produce all $T$ outputs | Space | Matrix type |
|---|---|---|---|---|
| Binary mech. | $1$ | $O(T)$ | $O(\log T)$ | sparse |
| Honaker Online | $0.5 + o(1)$ | $O(T \log T)$ | $O(\log T)$ | sparse |
| Denisov et al. | $0.0487\ldots + o(1)^{*}$ | $O(T^2)$ | $O(T^2)$ | dense |
| Henzinger et al. | $0.0487\ldots + o(1)$ | $O(T \log T)^{**}$ | $O(T)$ | Toeplitz |
| **Our mechanism** | $0.25 + o(1)$ | $O(T)$ | $O(\log T)$ | sparse |

*"Honaker Online" refers to the "Estimation from below" variant in Honaker (2015) as implemented in Kairouz et al. (2021). There is no explicit bound on variance in Denisov et al. (2022), but the method finds an optimal matrix factorization so it should achieve same variance $^{*}$ as Henzinger et al. (2023) up to lower order terms. As to their efficiency, Denisov et al. (2022) contains empirical work for reducing the time and space usage by approximating the matrices involved as a sum of a banded matrix and a low-rank approximation, but without formal guarantees. The time usage in $^{**}$ assumes implementing the matrix-vector product using FFT. Leveraging FFT for continual observation is not a novel approach (Choquette-Choo et al., 2023). All sparse matrices have $O(\log T)$ nonzero entries per row or column, and all dense matrices have $\Omega(T^2)$ nonzero entries — a Toeplitz matrix can be seen as intermediate in the sense of having $O(T)$ unique diagonals, allowing for efficient storage and multiplication.*

Aside from the natural interpretation of continual counting as the differentially private release of user statistics over time, mechanisms for continual counting (and more generally for releasing prefix sums) are used as a subroutine in many applications. Such a mechanism is for example used in Google's privacy-preserving federated next word prediction model (McMahan & Thakurta, 2022; Kairouz et al., 2021; Choquette-Choo et al., 2023), in non-interactive local learning (Smith et al., 2017), in stochastic convex optimization (Han et al., 2022) and in histogram estimation (Cardoso & Rogers, 2022; Chan et al., 2012; Huang et al., 2022; Upadhyay, 2019) among others.

Given the broad adoption of continual counting as a primitive, designing algorithms for continual counting that improve constants in the error while scaling well in time and space is of practical interest.

## 1.1 Our contributions

In this paper we introduce the *Smooth Binary Mechanism*, a differentially private algorithm for the continual counting problem that improves upon the original binary mechanism by Chan et al. (2011); Dwork et al. (2010) in several respects, formalized in Theorem 1.1 and compared to in Table 1.

**Theorem 1.1** (Smooth Binary Mechanism). *For any $\rho > 0$, $T > 1$, there is an efficient $\rho$-zCDP continual counting mechanism $\mathcal{M}$, that on receiving a binary stream of length $T$ satisfies*

$$\mathrm{Var}[\mathcal{M}(t)] = \frac{1 + o(1)}{8\rho}\log_2(T)^2$$

*where $\mathcal{M}(t)$ is the output prefix sum at time $t$, while only requiring $O(\log T)$ space, $O(T)$ time to output all $T$ prefix sums, and where the error is identically distributed for all $1 \le t \le T$.*

Our mechanism retains the scalability in time and space of the binary mechanism while offering an improvement in variance by a factor of $4 - o(1)$. It also has the *same* error distribution in every step by design, which could make downstream applications easier to analyze.

**Sketch of technical ideas.** Our starting point is the binary mechanism which, in a nutshell, uses a complete binary tree with $\ge T + 1$ leaves (first $T$ leaves corresponding to $x_1, \ldots, x_T$) in which each node contains the sum of the leaves below, made private by adding random noise (e.g. from a Gaussian distribution). To estimate a prefix sum $\sum_{i=1}^{t} x_i$ we follow the path from the root to the leaf storing $x_{t+1}$. Each time we go to a right child the sum stored in its sibling node is added to a counter. An observation, probably folklore, is that it suffices to store sums for nodes that are *left children*, so

suppose we do not store any sum in nodes that are right children. The number of terms added when computing the prefix sum is the number of 1s in the binary representation $\text{bin}(t)$ of $t$, which encodes the path to the leaf storing $x_{t+1}$. The *sensitivity* of the tree with respect to $x_t$, i.e., the number of node counts that change by 1 if $x_t$ changes, is the number of 0s in $\text{bin}(t-1)$. Our idea is to only use leaves that have *balanced* binary representation, i.e. same number of 0s and 1s (assuming the height $h$ of the tree is an even integer). To obtain $T$ useful leaves we need to make the tree slightly deeper — it turns out that height $h$ slightly more than $\log_2(T)$ suffices. This has the effect of making the sensitivity of every leaf $h/2$, and the noise in every prefix sum a sum of $h/2$ independent noise terms.

**Limitations.** As shown in Table 1 and Section 4, the smooth binary mechanism, while improving on the original binary mechanism, cannot achieve as low variance as matrix-based mechanisms such as Henzinger et al. (2023). However, given that the scaling of such methods can keep them from being used in practice, our variant of the binary mechanism has practical utility in large-scale settings.

## 1.2 Related work

All good methods for continual counting that we are aware of can be seen as instantiations of the *factorization mechanism* (Li et al., 2015), sometimes also referred to as the *matrix mechanism*. These methods perform a linear transformation of the data, given by a matrix $R$, then add noise according to the sensitivity of $Rx$, and finally obtain the sequence of estimates by another linear transformation given by a matrix $L$. To obtain unbiased estimates, the product $LRx$ needs to be equal to the vector of prefix sums, that is, $LR$ must be the lower triangular all 1s matrix.

Seminal works introducing the first variants of the binary mechanism are due to Chan, Shi, and Song (2011) and Dwork, Naor, Pitassi, and Rothblum (2010), but similar ideas were proposed independently in Hay, Rastogi, Miklau, and Suciu (2010) and Gehrke, Wang, and Xiao (2010). Honaker (2015) noticed that better estimators are possible by making use of *all* information in the tree associated with the binary tree method. A subset of their techniques can be leveraged to reduce the variance of the binary mechanism by up to a factor 2, at some cost of efficiency, as shown from their implementation in Kairouz, Mcmahan, Song, Thakkar, Thakurta, and Xu (2021).

A number of recent papers have studied improved choices for the matrices $L$, $R$. Denisov, McMahan, Rush, Smith, and Thakurta (2022) treated the problem of finding matrices leading to minimum largest variance on the estimates as a convex optimization problem, where (at least for $T$ up to 2048) it was feasible to solve it. To handle a larger number of time steps they consider a similar setting where a restriction to *banded* matrices makes the method scale better, empirically with good error, but no theoretical guarantees are provided.

Fichtenberger, Henzinger, and Upadhyay (2023) gave an explicit decomposition into lower triangular matrices, and analyzed its error in the $\ell_\infty$ metric. The matrices employed are Toeplitz (banded) matrices. Henzinger, Upadhyay, and Upadhyay (2023) analyzed the same decomposition with respect to mean squared error of the noise, and showed that it obtains the best possible error among matrix decompositions where $L$ and $R$ are square matrices, up to a factor $1 + o(1)$ where the $o(1)$ term vanishes when $T$ goes to infinity.

A breakdown of how our mechanism compares to existing ones is shown in Table 1. While not achieving as small an error as the factorization mechanism of Henzinger et al. (2023), its runtime and small memory footprint allows for better scaling for longer streams. For concreteness we consider privacy under $\rho$-zero-Concentrated Differential Privacy (zCDP) (Bun & Steinke, 2016), but all results can be expressed in terms of other notions of differential privacy.

## 2 Preliminaries

**Binary representation of numbers.** We will use the notation $\text{bin}(n)$ to refer to the binary representation of a number $n \in [2^h]$, where $h > 0$ is an integer, and let $\text{bin}(n)$ be padded with leading zeros to $h$ digits. For example, $\text{bin}(2) = \texttt{0b010}$ for a tree of height $h = 3$. When indexing such a number we let index $i$ refer to the $i^{th}$ least significant bit, e.g. $\text{bin}(2)_1 = 1$. We will also refer to prefixes and suffixes of binary strings, and we use the convention that a prefix of a binary string includes its most significant bits.

**Partial sums (p-sums).** To clear up the notation we use the concept of p-sums $\Sigma[i, j]$ where $\Sigma[i, j] = \sum_{t=i}^{j} x_t$. We will furthermore use the concept of noisy p-sums

$$\widehat{\Sigma}[i, j] = \Sigma[i, j] + X[i, j], \ X[i, j] \sim \mathcal{F}$$

where $\mathcal{F}$ is a suitable distribution for the DP paradigm, e.g. Laplacian or Gaussian. For convenience we also define $\hat{x}_t = \widehat{\Sigma}[t, t]$, i.e. $\hat{x}_t$ is the single stream element with noise added.

## 2.1 Continual observation of bit stream

Given an integer $T > 1$ we consider a finite length binary stream $x = (x_1, x_2, \ldots, x_T)$, where $x_t \in \{0, 1\}, 1 \leq t \leq T$, denotes the bit appearing in the stream at time $t$.

**Definition 2.1** (Continual Counting Query). Given a stream $x \in \{0, 1\}^T$, the *count* for the stream is a mapping $c : \{1, \ldots, T\} \to \mathbb{Z}$ such that $c(t) = \sum_{i=1}^{t} x_i$.

**Definition 2.2** (Counting Mechanism). A *counting mechanism* $\mathcal{M}$ takes a stream $x \in \{0, 1\}^T$ and produces a (possibly random) vector $\mathcal{M}_x \in \mathbb{R}^T$ where $(\mathcal{M}_x)_t$ is a function of the first $t$ elements of the stream. For convenience we will write $\mathcal{M}(t)$ for $(\mathcal{M}_x)_t$ when there is little chance for ambiguity.

To analyze a counting mechanism from the perspective of differential privacy, we also need a notion of neigboring streams.

**Definition 2.3** (Neighboring Streams). Streams $x, x' \in \{0, 1\}^T$ are said to be *neighboring*, denoted $x \sim x'$, if $|\{i \mid x_i \neq x'_i\}| = 1$.

Intuitively, for a counting mechanism to be useful at a given time $t$, we want it to minimize $|\mathcal{M}(t) - c(t)|$. We consider unbiased mechanisms that return the true counts in expectation and we focus on minimizing $\mathrm{Var}[\mathcal{M}(t) - c(t)]$.

## 2.2 Differential privacy

For a mechanism to be considered differentially private, we require that the outputs for any two neighboring inputs are are indistinguishable. We will state our results in terms of $\rho$-zCDP:

**Definition 2.4** (Concentrated Differential Privacy (zCDP) (Bun & Steinke, 2016)). For $\rho > 0$, a randomized algorithm $\mathcal{A} : \mathcal{X}^n \to \mathcal{Y}$ is $\rho$-zCDP if for any $\mathcal{D} \sim \mathcal{D}'$, $D_\alpha(\mathcal{A}(\mathcal{D})||\mathcal{A}(\mathcal{D}')) \leq \rho\alpha$ for all $\alpha > 1$, where $D_\alpha(\mathcal{A}(\mathcal{D})||\mathcal{A}(\mathcal{D}'))$ is the $\alpha$-Rényi divergence between $\mathcal{A}(\mathcal{D})$ and $\mathcal{A}(\mathcal{D}')$.

In the scenario where we are looking to release a real-valued function $f(\mathcal{D})$ taking values in $\mathbb{R}^d$, we can achieve zCDP by adding Gaussian noise calibrated to the $\ell_2$-sensitivity of $f$.

**Lemma 2.5** (Gaussian Mechanism (Bun & Steinke, 2016)). *Let $f : \mathcal{X}^n \to \mathbb{R}^d$ be a function with global $\ell_2$-sensitivity $\Delta := \max_{\mathcal{D} \sim \mathcal{D}'} \|f(\mathcal{D}) - f(\mathcal{D}')\|_2$. For a given data set $\mathcal{D} \in \mathcal{X}^n$, the mechanism that releases $f(\mathcal{D}) + \mathcal{N}(0, \frac{\Delta^2}{2\rho})^d$ satisfies $\rho$-zCDP.*

It is known that $\rho$-zCDP implies $(\rho + 2\sqrt{\rho \ln(1/\delta)}, \delta)$-differential privacy for every $\delta > 0$ (Bun & Steinke, 2016).

Lastly, when comparing counting mechanisms based on Gaussian noise, we note that it is sufficient to look at the variance. For a single output $\mathcal{M}(t)$, the variance $\mathrm{Var}[\mathcal{M}(t)]$ is the only relevant metric, as it completely describes the output distribution. For a series of outputs $\mathcal{M}_x$, and related norms $\|\mathcal{M}_x\|_p$, we note that a mechanism with lower variance will be more concentrated in each coordinate and have lower $p^{th}$ moment, allowing for tighter bounds on the norm.

## 2.3 Differentially private continual counting

We next describe two approaches to continual counting.

**Binary mechanism.** The binary mechanism (Chan et al., 2011; Dwork et al., 2010; Gehrke et al., 2010; Hay et al., 2010) can be considered the canonical mechanism for continual counting. In this section we present a variant of it where only left subtrees are used. The mechanism derives its name

from the fact that a binary tree is built from the input stream. Each element from the stream is assigned a leaf in the binary tree, and each non-leaf node in the tree represents a p-sum of all elements in descendant leaves. All values are stored noisily in nodes, and nodes are added together to produce a given prefix sum. Such a binary tree is illustrated in Figure 1(a).

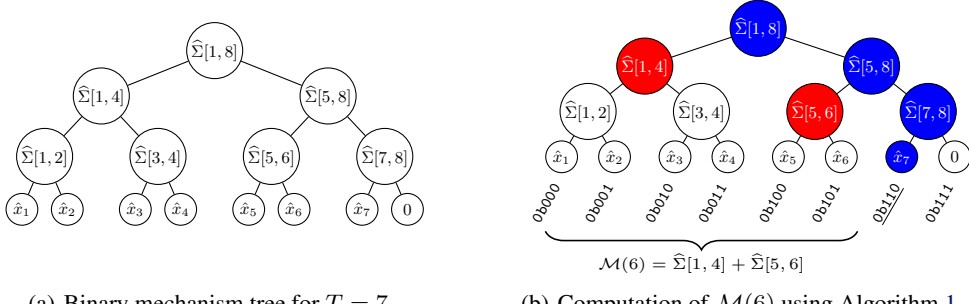

(a) Binary mechanism tree for $T = 7$.      (b) Computation of $\mathcal{M}(6)$ using Algorithm 1.

*Figure 1: Binary trees for a sequence of length $T = 7$. In Figure 1(b) each leaf is labeled by $\mathrm{bin}(t-1)$, and it illustrates how the prefix sum up to $t = 6$ can be computed from $\mathrm{bin}(t)$. Blue nodes describe the path taken by Algorithm 1, and the sum of red nodes form the desired output $\mathcal{M}(6)$.*

We will return for a closer analysis of the binary mechanism in Section 3.1. For now we settle for stating that the $\rho$-zCDP binary mechanism achieves $\mathrm{Var}[\mathcal{M}(t)] = O(\log(T)^2)$ and is known to be computationally efficient: to release all prefix sums up to a given time $t \leq T$ requires only $O(\log(T))$ space and $O(t)$ time.

**Factorization mechanism.** The binary mechanism belongs to a more general class of mechanisms called *factorization mechanisms* (Li et al., 2015), sometimes also referenced as *matrix mechanisms*. Computing a prefix sum is a linear operation on the input stream $x \in \mathbb{R}^T$, and computing all prefix sums up to a given time $T$ can therefore be represented by a matrix $A \in \mathbb{R}^{T \times T}$, where $c(t) = (Ax)_t$. $A$ is here a lower-triangular matrix of all 1s. The factorization mechanism characterizes solutions to the continual counting problem by factorizing $A$ as $A = LR$ with corresponding mechanism $\mathcal{M}_x = L(Rx + z)$, $z \sim \mathcal{F}^n$ where $n$ is the dimension of $Rx$.

Intuitively $Rx$ represents linear combinations of the stream, which are made private by adding noise $z$, and which then are aggregated by $L$. To achieve $\rho$-zCDP for this mechanism, we let $z \sim \mathcal{N}(0, \frac{\Delta^2}{2\rho})^n$ where $\Delta = \max_i \|Re_i\|_2$, $e_i$ being the $i^{th}$ unit vector. It follows that the corresponding output noise becomes $Lz$ with coordinates $(Lz)_i \sim \mathcal{N}(0, \frac{\Delta^2}{2\rho} \|L_i\|_2^2)$ where $L_i$ is the $i^{th}$ row in $L$.

**Extension to multidimensional input.** While we have so far assumed one-dimensional input, any one-dimensional $\rho$-zCDP factorization mechanism naturally generalizes to higher dimensions. This fact enables factorization mechanisms to be applied to the problem of private learning where gradients are summed. We sketch an argument for this fact in Appendix B.

## 3 Our mechanism

To introduce our variant of the binary mechanism, we need to return to the original mechanism.

### 3.1 Closer analysis of the binary mechanism

As has been pointed out in earlier work (Denisov et al., 2022; Henzinger et al., 2023), a naive implementation of the binary mechanism will lead to variance that is non-uniform with respect to time: the number of 1s in the bitwise representation of $t$ determines the variance. To underline this, consider the pseudo-code in Algorithm 1 for computing a prefix sum given a binary mechanism tree structure. The code assumes that the tree has $\geq t + 1$ leaves, a detail that only matters when $t$ is a power of 2 and allows us to never use the root node. To illustrate a simple case, see Figure 1(b).

---
**Algorithm 1** Prefix Sum for Binary Mechanism
---
1: **Input:** binary tree of height $h$ storing $\widehat{\Sigma}[a, b]$ for $b \leq t$, time $t \leq 2^h - 1$
2: $output \leftarrow 0$
3: $s \leftarrow \text{bin}(t)$ {padded to $h$ bits by adding zeros}
4: $a \leftarrow 1; b \leftarrow 2^h$
5: **for** $i = h - 1$ **to** $0$ **do**
6:     $d = \lfloor \frac{a+b}{2} \rfloor$
7:     **if** $s_i = 1$ **then**
8:         $output \leftarrow output + \widehat{\Sigma}[a, d]$
9:         $a \leftarrow d + 1$
10:    **else**
11:        $b \leftarrow d$
12:    **end if**
13: **end for**
14: **return** $output$
---

We can make the following two observatons:

- $\text{bin}(t - 1)$ encodes a path from the root to the leaf where $x_t$ is stored.
- To compute prefix sums using Algorithm 1, we only need to store values in "left children".

Combining these observations, we get the following result:

**Proposition 3.1.** *The squared $\ell_2$-sensitivity $\Delta^2$ of $x_t$ is equal to the number of 0s in $\text{bin}(t - 1)$.*

To see this, note that the number of 0s in $\text{bin}(t - 1)$ is equal to the number of left-children that are passed through to reach the given node. Changing the value of $x_t$ impacts all its ancestors, and since only left-children are used for prefix sums, the result immediately follows. This does not address the volatility of variance with respect to time. However, studying Algorithm 1 gives the reason:

**Proposition 3.2.** $\text{Var}[\mathcal{M}(t)]$ *is proportional to the number of 1s in* $\text{bin}(t)$.

The result follows from the fact that the number of terms added together for a given prefix sum $c(t)$ is equal to the number of 1s in $\text{bin}(t)$, since Line 8 is executed for each such 1. In this view, each node that is used for the prefix sum at time $t$ can be identified as a prefix string of $\text{bin}(t)$ that ends with a 1.

We will return to Proposition 3.1 and Proposition 3.2 when constructing our smooth mechanism, but for now we settle for stating that combined they give the exact variance at each time step for the binary mechanism. Following Li et al. (2015), to make the mechanism private we have to accommodate for the worst sensitivity across all leaves, which yields Theorem 3.3.

**Theorem 3.3** (Exact Variance for Binary Mechanism (Chan et al., 2011; Dwork et al., 2010))**.** *For any $\rho > 0$, $T > 1$, the $\rho$-zCDP binary mechanism $\mathcal{M}$ based on Algorithm 1 achieves variance*

$$\text{Var}[\mathcal{M}(t)] = \frac{\lceil \log(T + 1) \rceil}{2\rho} \| \text{bin}(t) \|_1$$

*for all $1 \leq t \leq T$, where $\| \text{bin}(t) \|_1$ is equal to the number of 1s in* $\text{bin}(t)$.

### 3.2 A smooth binary mechanism

Based on the analysis, a naive idea to improve the binary mechanism would be to only consider leaves with "favorable" time indices. To make this a bit more precise, we ask the following question: could there be a better mechanism in which we store elements in only a subset of the leaves in the original binary tree, and then use Algorithm 1 to compute the prefix sums? We give an affirmative answer.

Consider a full binary tree of height $h$ where we let the leaves be indexed by $i$ (1-indexed). Given a sequence of elements $x \in \{0, 1\}^T$ (we assume a great enough height $h$ to accommodate for all elements) and an integer $0 \leq k \leq h$, we conceptually do the following for each leaf with index $i$:

- If $\text{bin}(i - 1)$ has $k$ 0s, store the next element of the stream in the leaf.
- Otherwise store a token 0 in the leaf.

More rigorously stated, we are introducing an injective mapping from time to leaf-indices $m$ : $\{1, \ldots, T + 1\} \to [2^h + 1]$, such that $m(t)$ is the $(t-1)^{st}$ smallest $h$-bit integer with $k$ 0s in its binary representation. $x_t$ gets stored in the leaf with index $m(t)$, and to compute $\mathcal{M}(t)$, we would add p-sums based on $\text{bin}(m(t+1))$. The resulting algorithm is listed as Algorithm 2, where $\widehat{\Sigma}[i, j]$ is the noisy count of leaf $i$ through $j$.

---

**Algorithm 2** Prefix Sum for Smooth Binary Mechanism

1: **Input:** binary tree of height $h$ storing $\widehat{\Sigma}[a, b]$ for $b \leq m(t)$, time $t$ where $m(t+1) \leq 2^h$
2: $output \leftarrow 0$
3: $s \leftarrow \text{bin}(m(t+1))$ {padded to $h$ bits by adding zeros}
4: $a \leftarrow 1; b \leftarrow 2^h$
5: **for** $i = h - 1$ **to** 0 **do**
6:    $d = \lfloor \frac{a+b}{2} \rfloor$
7:    **if** $s_i = 1$ **then**
8:       $output \leftarrow output + \widehat{\Sigma}[a, d]$
9:       $a \leftarrow d + 1$
10:   **else**
11:      $b \leftarrow d$
12:   **end if**
13: **end for**
14: **return** $output$

---

It follows that the $\ell_2$-sensitivity is equal to $\sqrt{k}$, and that any prefix sum will be a sum of $h - k$ nodes. Importantly, the latter fact removes the dependence on $t$ for the variance.

**Choosing a k.** The optimal choice of $k$ depends on the differential privacy paradigm. Here we only consider $\rho$-zCDP where $\Delta = \sqrt{k}$. Since each prefix sum is computed as a sum of $h - k$ nodes, the variance for a given prefix sum becomes $\text{Var}[\mathcal{M}(t)] = (h - k) \cdot \frac{k}{2\rho}$, which for $k = h/2$ gives a leading constant of $1/4$ compared to the maximum variance of the binary mechanism. This choice of $k$ is valid if the tree has an even height $h$, and it maximizes the number of available leaves. Such a tree together with a computation is shown in Figure 2.

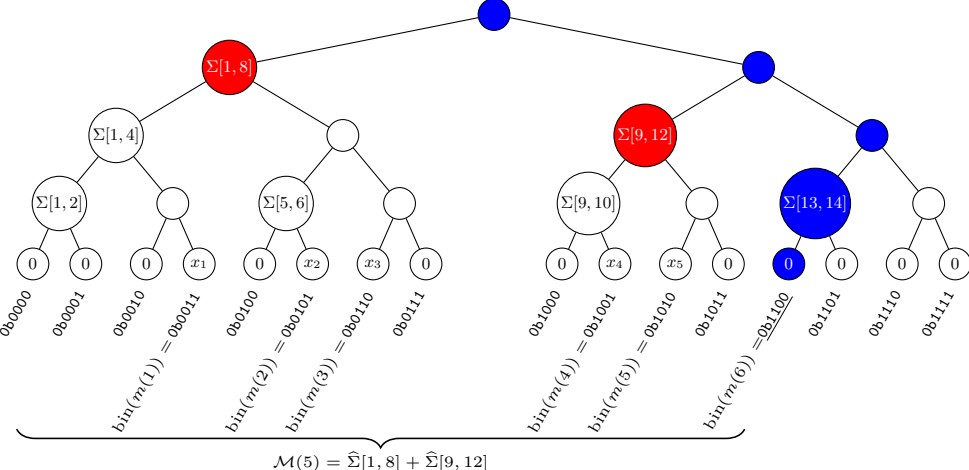

$$\mathcal{M}(5) = \widehat{\Sigma}[1, 8] + \widehat{\Sigma}[9, 12]$$

*Figure 2: Computation of $\mathcal{M}(5)$ using the smooth binary mechanism where $T = 5$. The figure illustrates how the prefix sum up to $t$ can be computed from $\text{bin}(m(t+1))$. Blue nodes describe the path taken by Algorithm 2, and the sum of red nodes form the desired output $\mathcal{M}(t)$. All values shown in nodes are stored noisily, and $\Sigma[i, j]$ is here defined as the sum of leaves $i$ through $j$. Observe that the noise in each node is drawn from the same distribution and that each query is formed by adding together $h/2 = 2$ noisy nodes, implying an identical distribution of the error at each step.*

**A penalty in height.**  The analysis above assumes that we have a tree of sufficient height. If we before had a tree of height $\lceil \log(T+1) \rceil$, we now need a tree of height $h \geq \lceil \log(T+1) \rceil$ to have enough leaves with the right ratio of 1s in their index. To account for this, we let $h$ be the smallest even integer such that $h \geq \lceil \log(T) \rceil + a \log \log T$, where $a$ is a constant. For our new tree to support as many prefix sums, we need that $\binom{h}{h/2} \geq T+1$. This holds for $a > 1/2$ and sufficiently large $T$, but we show it for $a = 1$ next. Using Stirling's approximation in the first step, we can establish that

$$\binom{h}{h/2} \geq \frac{2^h}{\sqrt{2h}} \geq \frac{2^{\log T + \log \log T}}{\sqrt{2}\sqrt{\lceil \log T \rceil + \log \log T}} = \frac{\log T}{\sqrt{2}\sqrt{\lceil \log T \rceil + \log \log T}} \cdot T \ ,$$

which is at least $T+1$ for $T \geq 13$.

**Resulting variance.**  This height penalty makes $\mathrm{Var}[\mathcal{M}(t)]$ no longer scale as $\log(T+1)^2$, but $(\log(T) + a \log \log(T))^2$. Nevertheless, we can state the following:

**Lemma 3.4.**  *For any $\rho > 0$, $T \geq 13$, the $\rho$-zCDP smooth binary mechanism $\mathcal{M}$ achieves variance*

$$\mathrm{Var}[\mathcal{M}(t)] = \frac{1 + o(1)}{8\rho} \log(T)^2$$

*where $1 \leq t \leq T$, and the $o(1)$ term is at most $\frac{2 \log \log(T)}{\log(T)} + \left(\frac{\log \log(T)}{\log(T)}\right)^2$.*

which is an improvement over the original binary mechanism by a factor of $1/4$ with regard to the leading term. This improvement is shown empirically in Section 4.

**Constant average time per output.**  When outputting $T$ prefix sums continuously while reading a stream, we only have to store the noise of the nodes, not the p-sums themselves. To make this more explicit, let $S_t$ describe the set of nodes (p-sum indices) that the smooth mechanism adds together to produce the output at time $t$. To produce $\mathcal{M}(t+1)$ given $\mathcal{M}(t)$, we effectively do:

$$\mathcal{M}(t+1) = \mathcal{M}(t) + x_{t+1} + \sum_{(i,j) \in S_{t+1} \setminus S_t} X[i,j] - \sum_{(i,j) \in S_t \setminus S_{t+1}} X[i,j] \ .$$

To quantify the cost, we need to deduce how many nodes are replaced from $t$ to $t+1$, which means reasoning about $S_t$ and $S_{t+1}$. Recalling that each element in $S_t$ can be identified by a prefix string of $\mathrm{bin}(m(t+1))$ terminating with a 1, consider Figure 3. Based on the pattern shown in Figure 3, where the leaf indices only differ in their least significant bits, we get that $|S_{t+1} \setminus S_t| = |S_t \setminus S_{t+1}| = n$, where $n$ is the number of 1s in the least significant block of 1s. We formalize this observation next to give a bound on the average cost when outputting a sequence of prefix sums.

**Lemma 3.5.**  *Assuming the cost of replacing a node in a prefix sum is 1, then the cost to release all $\binom{2k}{k} - 1$ prefix sums in the tree of height $2k$ using the smooth binary mechanism is at most $2\binom{2k}{k}$.*

*Proof.*  As argued before, to compute $\mathcal{M}(t+1)$ given $\mathcal{M}(t)$ we need to replace a number of nodes equal to the size $n$ of the least significant block of 1s in $\mathrm{bin}(m(t+1))$. We can therefore directly compute the total cost by enumerating all valid indices with different block sizes as

$$\mathrm{cost} = -k + \sum_{n=1}^{k} \sum_{i=k-n}^{2k-n-1} n\binom{i}{k-n} = -k + \sum_{n=1}^{k} n\binom{2k-n}{k-n+1}$$

$$= -k + \sum_{i=1}^{k} \sum_{j=1}^{i} \binom{k-1+j}{k-1} = -2k + \sum_{i=1}^{k} \binom{k+i}{k} = \binom{2k+1}{k+1} - (2k+1) \leq 2\binom{2k}{k}$$

where the initial $-k$ term comes from excluding the last balanced leaf index in the tree.  $\square$

Figure 3: *The least significant bits of two leaf indices in a full binary tree that are neighboring with respect to time when used in the smooth binary mechanism. If the first cluster of 1s in $\mathrm{bin}(m(t+1))$, counted from the least significant bit, has $n$ 1s then $n$ nodes in total will be replaced from $t$ to $t+1$.*

In particular, this implies that the average cost when releasing all prefix sums in a full tree is $\leq 2$. For $T$ that does not use all leaves of a tree, comparing to the closest $T'$ where $T \leq T' = \binom{2k}{k} - 1$ also implies an average cost of $\leq 4$ for arbitrary $T$. For a single output the cost is $O(\log(T))$. It is not hard to check that computing $\text{bin}(m(t+1))$ from $\text{bin}(m(t))$ can be done in constant time using standard arithmetic and bitwise operations on machine words.

**Logarithmic space.** The argument for the binary mechanism only needing $O(\log(T))$ space extends to our smooth variant. We state this next in Lemma 3.6, and supply a proof in the appendix.

**Lemma 3.6.** *The smooth binary mechanism computing prefix sums runs in $O(\log(T))$ space.*

**Finishing the proof for Theorem 1.1.** The last, and more subtle, property of our mechanism is that the error at each time step not only has constant variance, but it is identically distributed. By fixing the number of 1s in each leaf index to $k$, the output error at each step becomes the $k$-fold convolution of the noise distribution used in each node, independently of what that distribution is. Our mechanism achieves this by design, but note that this can be achieved for any other mechanism (e.g. the regular binary mechanism) by adding fresh noise in excess of what gives privacy (at the expense of utility). Combining Lemmas 3.4, 3.5 and 3.6 together with this last property, we arrive at Theorem 1.1.

## 4 Comparison of mechanisms

In this section we review how the smooth binary mechanism compares to the original binary mechanism, and the factorization mechanism of Henzinger et al. (2023). A Python implementation of our smooth binary mechanism (and the classic binary mechanism) can be found on https://github.com/jodander/smooth-binary-mechanism. We do not compare to Denisov et al. (2022) since their method is similar to that of Henzinger et al. (2023) in terms of error, and less efficient in terms of time and space usage. We also do not compare to the online version of Honaker (2015) as it is less efficient in time, and its improvement in variance over the binary mechanism is at most a factor of 2, and can therefore be estimated from the figures.

**Variance comparison.** To demonstrate how the variance behaves over time for our smooth binary mechanism, see Figure 4. Given a fix $T$, the tree-based mechanisms compute the required tree height

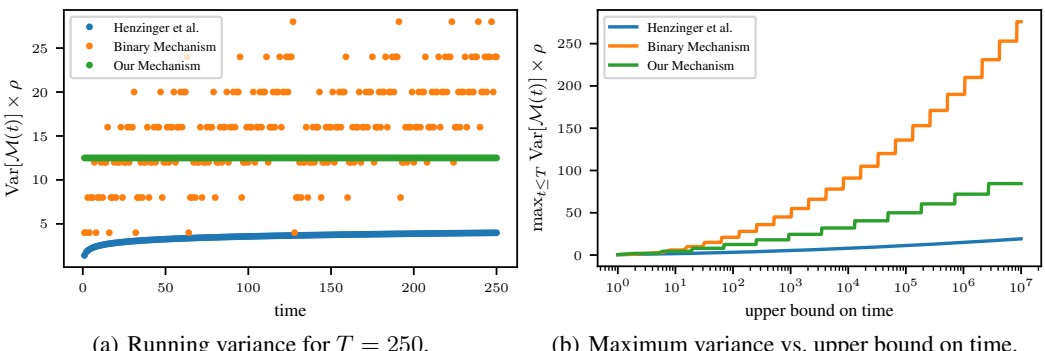

(a) Running variance for $T = 250$.  (b) Maximum variance vs. upper bound on time.

*Figure 4: Comparison of variance between the mechanism in Henzinger et al. (2023), the standard binary mechanism and our mechanism. Figure 4(a) shows $\text{Var}[\mathcal{M}(t)]$ for $1 \leq t \leq T$ for $T = 250$, whereas Figure 4(b) shows the maximum variance that each mechanism would attain for a given upper bound on time. At the last time step in Figure 4(b), our mechanism reduces the variance by a factor of $3.27$ versus the binary mechanism.*

to support all elements, and the factorization mechanism a sufficiently large matrix. The volatility of the error in the regular binary mechanism is contrasted by the stable noise distribution of our smooth binary mechanism, as demonstrated in Figure 4(a). In terms of achieving the lowest variance, the factorization mechanism in Henzinger et al. (2023) is superior, as expected. This result is replicated in Figure 4(b) where our mechanism offers a substantial improvement in terms of maximum variance over the original binary mechanism, but does not improve on Henzinger et al. (2023).

**Computational efficiency comparison.** While our mechanism does not achieve as low noise as the mechanism of Henzinger et al. (2023), it scales well in time and space, and with respect to the dimensionality of the stream elements. This is empirically demonstrated in Figure 5. Since the matrix

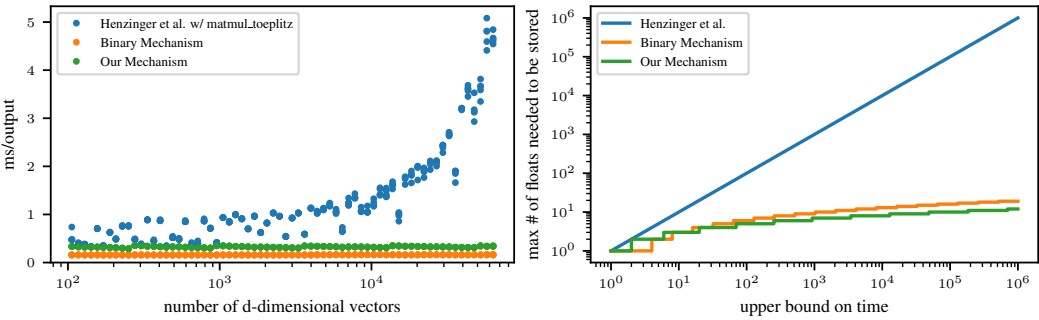

(a) Average computational time per output.  (b) Maximum space needed vs upper bounds on time.

*Figure 5: Comparison of computational efficiency between the mechanism in Henzinger et al. (2023), the binary mechanism and our mechanism. Figure 5(a) shows the computation time spent per $d$-dimensional input. The simulation was run $5$ times for each method, meaning each method has $5$ data points in the plot per time step. The computation was performed for elements of dimension $d = 10^4$, was run on a Macbook Pro 2021 with Apple M1 Pro chip and 16 GB memory using Python 3.9.6, scipy version 1.9.2, and numpy version 1.23.3. Figure 5(b) shows the maximum number of floats that has to be stored in memory when outputting all prefix sums up to a given time, assuming binary input.*

used in Henzinger et al. (2023) is a Toeplitz matrix, the scipy method "matmul_toeplitz" (based on FFT and running in time $O(dT \log T)$ to produce a $d$-dimensional output) was used to speed up the matrix multiplication generating the noise in Figure 5(a).

The discrepancy in the computation time scaling can largely be attributed to space: the needed space for these tree-based methods scales logarithmically with $T$, and linearly with $T$ for matrix-multiplication based methods. This is demonstrated in Figure 5(b). As to the difference in computation time between the tree-based methods, the smooth binary mechanism generates twice as much fresh noise per time step on average, which likely is the dominating time sink in this setup.

## 5   Conclusion and discussion

We presented an improved "smooth" binary mechanism that retains the low time complexity and space usage while improving the additive error and achieving stable noise at each time step. Our mechanism was derived by performing a careful analysis of the original binary mechanism, and specifically the influence of the binary representation of leaf indices in the induced binary tree. Our empirical results demonstrate the stability of the noise and its improved variance compared to the binary mechanism. The factorization mechanism of Henzinger et al. (2023) offers better variance, but is difficult to scale to a large number of time steps, especially if we need high-dimensional noise.

We note that the smooth binary mechanism can be extended to $\varepsilon$-DP. The optimal fraction of 1s in the leaves of the binary tree would no longer be $1/2$. An interesting problem is to find mechanisms that have lower variance and attractive computational properties. It is possible that the dependence on $T$ can be improved by leaving the factorization mechanism framework, but in absence of such a result the best we can hope is to match the variance obtained by Henzinger et al. (2023).

## Acknowledgments and disclosure of funding

We would like to thank Jalaj Upadhyay for useful discussions about our results and their relation to previous work. We would also like to thank the reviewers for their comments and suggestions which have contributed to a clearer paper. The authors are affiliated with Basic Algorithms Research Copenhagen (BARC), supported by the VILLUM Foundation grant 16582, and are also supported by Providentia, a Data Science Distinguished Investigator grant from Novo Nordisk Fonden.

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

# A    Proof of Lemma 3.6

The proof of Lemma 3.6 was omitted from the main text, it is given next.

*Proof.* The critical observation to make is that once a given p-sum is removed from a prefix sum, then it will never re-appear. Let its associated prefix string at time $t$ be $s$ of length $l$. If we look at the same $l$ bit positions in $\operatorname{bin}(m(t))$ as $t$ increases and interpret it as a number, then it is monotonously increasing with $t$. This implies that once a given prefix-string $s$ in $\operatorname{bin}(m(t))$ disappears at $t' > t$ then it will not be encountered again. We can therefore free up the memory used for storing any p-sum the moment it is no longer used. Because of this, it suffices to only store the p-sums in the active prefix sum at any time, of which there are at most $O(\log(T))$, proving the statement. $\qquad\square$

# B    Factorization mechanisms on multidimensional input

Assuming a $d$-dimensional input stream with elements $x_i \in \mathbb{R}^d$, let two streams $x, x'$ be neigboring if they differ at exactly one time step $t$ where $\|x_t - x'_t\|_2 \leq 1$. Given a one-dimensional $\rho$-zCDP factorization mechanism for continual counting with factorization $A = LR$, we want to argue that running it along each dimension separately gives a $\rho$-zCDP factorization mechanism for releasing prefix sums on $x$.

To see this, consider the new, flattened vector input $\tilde{x} \in \mathbb{R}^{d \cdot T}$ where $\tilde{x} = [(x_1)_1, (x_1)_2, \ldots, (x_1)_d, (x_2)_1, \ldots, (x_T)_d]$, i.e., $(x_t)_j = \tilde{x}_{(t-1) \cdot d + j}$. Analogously for these new inputs, we can define a counting matrix $\tilde{A} = A \oplus I_{d \times d} \in \mathbb{R}^{d \cdot T \times d \cdot T}$ that sums each dimension separately, where $\oplus$ is the Kronecker product and $I_{d \times d}$ is the $d$-dimensional identity matrix. We get a corresponding factorization $\tilde{A} = \tilde{L}\tilde{R}$ where $\tilde{L} = L \oplus I_{d \times d}$ and $\tilde{R} = R \oplus I_{d \times d}$.

It follows immediately that, $\tilde{L}, \tilde{R}$ gives a factorization mechanism for privately releasing $\tilde{A}\tilde{x}$, and from it we can extract all private prefix sums on the original stream $x$. Letting $\tilde{x}'$ be the vector representation of the stream $x' \sim x$, we will reason about the $\ell_2$-sensitivity $\Delta$ next. First observe that for neighbouring inputs $x$ and $x'$ that differ at time $t$, we have:

$$\left\| \tilde{R}(\tilde{x} - \tilde{x}') \right\|_2 = \|Re_t\|_2 \cdot \|\tilde{x} - \tilde{x}'\|_2$$

and therefore

$$\Delta = \sup_{\tilde{x} \sim \tilde{x}'} \left\| \tilde{R}(\tilde{x} - \tilde{x}') \right\|_2 = \max_i \|Re_i\|_2 \cdot \sup_{\tilde{x} \sim \tilde{x}'} \|\tilde{x} - \tilde{x}'\|_2 = \max_i \|Re_i\|_2 \ ,$$

which equates the $\ell_2$-sensitivity of the one-dimensional case. Recapitulating, any one-dimensional $\rho$-zCDP factorization mechanism for continual counting is also a $\rho$-zCDP counting mechanism for inputs of higher dimension, as long as the neigboring relation is appropriately extended.

