# OpenReview forum: "A Smooth Binary Mechanism for Efficient Private Continual Observation"
_NeurIPS.cc/2023/Conference — NeurIPS 2023 poster_

### Official Review · Reviewer_pQUM · 2023-06-20

**Soundness:** 4 excellent
**Presentation:** 3 good
**Contribution:** 3 good
**Rating:** 6
**Confidence:** 4

**Summary:**

The binary mechanism is a simple, efficient and well-known method to answer linear prefix queries (like counting) when a database of size T is not given in advanced, but rather in each time point $t \in [T]$ we are given a new row $x_t$, and we are required to update our result based on the new row (while taking into account that the previous estimations were already released). For the prefix sum (or counting) problem where each row is 0 or 1, the binary mechanism in the zCDP model results with an additive error of Gaussian noise for each observation, where the maximal noise variance is $0.5 \cdot \log_2^2 T$. The noise variance is not identical for every observation, as it is a function of the number of zeros in the binary representation of $t-1$ (the observation index). This can also be change to be depend on the number of ones instead of the number of zeros.
The main contribution of this work is to develop a (simple but clever) variant of the binary mechanism that balances between the different noises such that each observation will get the same noise variance, and this variance will be only $(0.125 + o(1)) \cdot \log_2^2 T$, which is almost $4$ times better than the maximal variance of the binary mechanism (which translates to factor $2$ in the resulting noise magnitude).
This work does not improve upon the best-known variance, as Henzinger et al. (SODA 2023) and Denisov et al. (NeurIPS 2022) have managed to get variance of $0.04865 \cdot \log_2^2 T$, which is $10.2$ times better than the standard binary mechanism and $2.57$ times better than this work. However, these works are much more complex, and more importantly, much less efficient in terms of running time and space.


**Strengths:**

At the bottom line, this paper improves a factor of $4$ in the variance of the Binary Mechanism with almost no additional cost in its complexity. Therefore, it potentially can be useful in practice. It does not achieve the best accuracy, but definitely provides a simpler and more efficient alternative to the more accurate but complex algorithms of Henzinger et al. (SODA 2023) and Denisov et al. (NeurIPS 2022).

**Weaknesses:**

The improvement is clever and simple, but perhaps too simple for a NeurIPS submission. I lean towards accepting this paper mainly because it might be useful in practice.


**Questions:**

No questions to the authors.

**Limitations:**

Yes

---

> ### Author Rebuttal · Authors · 2023-08-09
>
> We thank the reviewer for their reading and agree that the algorithm has the potential of being useful in practice.
>
> *W1: The improvement is clever and simple, but perhaps too simple for a NeurIPS submission.*
>
> While we agree that our algorithm is simple, we would argue that this is a strength of the algorithm rather than a weakness. It being a simple algorithm may come as a surprise given that no streaming algorithm using sublinear space for the problem has achieved a constant improvement in the variance since the seminal binary mechanism (Chan et al., 2011; Dwork et al., 2010).
>
> The question of whether there are algorithms for continual counting that combine the efficiency of the binary mechanism with a lower variance has also been raised earlier. It was for example raised during a presentation of the work (Denisov et al., 2022) at a Google workshop last year.

---

> > ### Comment · Reviewer_pQUM · 2023-08-16
> >
> > Thank you for the rebuttal.
> >
> > I was not aware of this question before (i.e., to come up with an efficient algorithm as the binary mechanism but with a lower variance). From reading your paper, I was slightly surprised that such a variant was not considered before as it seems very natural to try to balance the noises as you do. But perhaps I feel like that just because you managed to explain the limitations of the binary mechanism pretty well, and you gave a very intuitive description of your algorithm.
> >
> > I'd like to add my opinion regarding your writing after reading the comments of the other reviewers.
> > I agree with them that the paper feels not formal enough, (in particular, your algorithm should be fully described in the main body).
> > However, I chose not to write it as a weakness, because in the end I did manage to understand exactly what you are doing even without the formal descriptions, and I like the way you chose the present it. Usually, I don't like papers that are not formal enough, but because your solution is really simple and intuitive, then it felt ok.

---

> > > ### Author Response · Authors · 2023-08-17
> > >
> > > Thank you to the reviewer for their comments.
> > >
> > > To improve the exposition, we will dedicate additional space in the main body of the revised paper to describing our algorithm. This includes moving Algorithm 2 (the full formal specification of the final algorithm) from the Appendix to the main body, and adding the figure(s) shown in our global rebuttal response.

---

### Official Review · Reviewer_aGJ1 · 2023-06-24

**Soundness:** 3 good
**Presentation:** 3 good
**Contribution:** 3 good
**Rating:** 7
**Confidence:** 4

**Summary:**

This paper studies the continual counting with differential privacy problem. Here there are $T$ time steps; at time step $t \\in [T]$, we receive a $x_t \\in \\{0, 1\\}$ and we have to output an estimate to the sum so far $x_1 + \\cdots + x_t$. We would like to entire stream of output to be differentially private (DP) w.r.t. changing a single entry $x_t$. Below we will discuss the privacy guarantees in terms of concentrated DP (CDP) but this can be easily translated to standard approximate-DP guarantee.

The classic binary tree mechanism [Chan et al., 2011; Dwork et al., 2010] solves this problem by building a complete binary tree with $T$ leaves, each representing $x_t$ for $t \in [T]$. Each node in the tree represents the sum of all leaves below. Appropriate noise are added to each node to make sure this data structure is private. When we want to output an estimate, we take the sum of the appropriate nodes; it is not hard to see that then number of nodes required are at most $O(\\log T)$. This gives error (standard deviation) for each estimate that is $O(\\log T)$, a significant improvement over the trivial algorithm (that release each estimate with independent noise)--which has error $O(\sqrt{T})$. The binary tree mechanism belongs to a class of mechanism called matrix mechanism based on a factorization of the "workload" matrix, which is the lower triangular all-1 matrix in the continual counting case. [Denisov et al., 2022] experimented with matrix mechanism and showed significant error improvement (more than 3x) over the binary tree mechanism. [Henzinger et al., 2023] gave an elegant factorization based on Toeplitz matrices and prove that it is asymptotically optimal.

Despite the (asymptotic) optimality of [Denisov et al., 2022, Henzinger et al., 2023], the algorithms are harder to implement than the binary tree mechanism. In particular, the noise generation requires $O(T)$ time and $O(T)$ space (for each update), whereas the binary tree mechanism (by keeping the noise only of the "active" node) requires only $O(1)$ amortized time (or $O(\\log T)$ worst case time) and $O(T)$ space. On the other hand, as stated above, the error of the binary tree mechanism is more than 3x larger than those of [Denisov et al., 2022, Henzinger et al., 2023]. This brings us to the contribution of this work: **the authors give an algorithm with the same time / space complexity as binary tree mechanism but with error reduced by a half** (so it is now only ~1.6x more than [Denisov et al., 2022, Henzinger et al., 2023]).

To understand the techniques, we need to discuss the binary tree mechanism in more detail. There are two important parameters that affect the error:
1. *Sensitivity*: This is the amount of contribution from each $x_t$ to the datastructure (measured in $\ell_2$ norm), which directly determines the scale of the noise needed to provide privacy.
2. *Number of nodes to combine*: This is the number of nodes that needs to be summed together to produce an estimate; the larger this number is, the larger number of noises get added together--leading to a larger error.

For 1., the observation (which is perhaps folklore) to start with is that we never use the right children when computing the estimates anyway, so we can remove these from the trees. This leaves us with an interesting situation: each $x_t$ does not have the same sensitivity. In particular, the number of nodes $x_t$ contributes to is now the number zeros in the binary representation of $t - 1$. For 2., it can be seen that different estimates requires adding up different number of nodes: specifically, the number of nodes needed for $x_1 + \\cdots + x_t$ is the number of ones in the binary representation of $t$. This leads to a rather natural and elegant strategy that the authors propose: instead of using all leaves, only use those that have equal number of zeros and ones in their binary representations. This reduces the sensitivity by $\\sqrt{2}$ and reduces the number of node to combined by a factor of $2$, leading to a total of $2$ saving in the standard deviation of the estimates compared to the vanilla binary tree mechanism. Note that since there are only $\\Theta(1/\\sqrt{T})$ fraction of leaves with equal zero-one in their binary representations, we actually have to look at a slightly higher tree than before, but this only contributes to $O(\log \log T)$--which is a lower order term--in the error.

## Post-Rebuttal Comments

Thank you the authors for the responses. As stated in my reply, I'd still strongly encourage the authors to clarify W2 and Q1 in more detail in the revision.

**Strengths:**

- Continual counting is a basic problem in literature and this paper provides a significant improvement over the classic binary tree mechanism.

- The approach is very elegant is likely practical.

**Weaknesses:**

- Although this paper's algorithm is more efficient than those in [Denisov et al., 2022, Henzinger et al., 2023], the error is still quite noticeably larger (by ~1.6x factor) for the same privacy level. Therefore, this is a tradeoff between time-space efficiency and privacy-utility tradeoff. In my opinion, the latter is typically more important in practice.

- The writing is not ideal:
  - Main algorithm (Algorithm 2) is not presented in the main body.
  - Despite discussing and using the high-dimensional version (where each $x_i$ is a $d$-dimensional vector) in the experiments, the main proof sections only consider the case where $x_i$ is a binary scalar.
  - "Streaming support" and "same error distribution" is discussed in a possibly misleading manner (see the first two questions below in more detail).

**Questions:**

- Line 57-58: "Unlike existing mechanisms it has the same error distribution in every step, which could make downstream applications easier to analyze." I am not sure why this is a big deal, since the noise distribution is just a Gaussian we can always just add another independent Gaussian to make it the same, right? (For example, if at the $t$-time step, my variance is $\\sigma_t^2$ whereas my worst case variance is $\\sigma_{\\max}^2$, then I can just add a noise drawn from $\\mathcal{N}(0, \\sigma_{\\max}^2 - \\sigma_t^2)$ to get the error distribution to be the same, right?)

- In Table 1, it is said that if we precompute the noise for Henzinger et al.'s algorithm, then we would not be able to support streaming setting. I do not understand why this is the case. Can you please elaborate? (Note that we have to be talking about fix time horizon $T$ anyway, since even your algorithm would need this to be able to set the noise magnitude.)

- Since Henzinger et al.'s factorization is pretty explicit (and has very nice form), is there any (informal) reason why one should think that it is hard to implement their algorithm using say polylogarithmic space and time?

- It seems like for your algorithm we only need the binary representation of every contributed time step minus one to have *at most* $h/2$ zeros and the binary representation of every queried time step to have *at most* $h/2$ ones. Since about half of the leaves have at most $h/2$ zeros (or $h/2$ ones), can you use this, instead of *exactly* $h/2$ to reduce the extra height to just $O(1)$ instead of the current $O(\\log \\log T)$? (Note that I think you do not need the contributed / queried leaves to be exactly equal; you only need a sequence $t_1, \dots, t_T$ for contribution and $q_1, \dots, q_T$ for queries that are interleaved, i.e. $t_1 \leq q_1 < t_2 \leq q_2 < \cdots < t_T \leq q_T$.)

## Minor comments

Below are minor comments to help the authors improve their revision. Please do *not* reply to them during rebuttal.

- Line 49: "algoritm" -> "algorithm"

- Line 91 - 94: I believe the fact that finding the optimal matrix mechanism can be formulated as a convex optimization problem was well known before [Denisov et al.]. See for example, the paper "Factorization Norms and Hereditary Discrepancy" by Matousek et al. and references therein.

---

> ### Author Rebuttal · Authors · 2023-08-09
>
> We thank the reviewer for their careful reading and insightful questions.
>
> *W1: [...] a tradeoff between time-space efficiency and privacy-utility tradeoff. In my opinion, the latter is typically more important in practice.*
>
> While often true, time-space efficiency is the reason why papers based on matrix mechanisms rarely go beyond $T=10^6$, and efficiency is in fact a real consideration for large-scale applications, e.g. private learning.
>
> *W2: No analogous proof for when $x_i$ is multi-dimensional despite showing such an experiment.*
>
> While true, the feasibility of the algorithm for this case follows trivially by composition.
>
> *Q1: Why is it a big deal that the error at each step is identically distributed when for a Gaussian mechanism this can always be achieved by adding extra fresh noise to match the worst case variance?*
>
> This is indeed correct for Gaussian mechanisms, and any other mechanism based on a stable distribution, and will be further emphasized in the paper. For mechanisms based on non-stable distributions it is non-trivial to achieve this property. In particular, if we want pure DP, then using our mechanism with Laplace noise will yield a pure DP mechanism where the error at each step is identically distributed.
>
> *Q2: Why does the precomputation of noise render Henzinger et al.'s algorithm a non-streaming algorithm?*
>
> Good observation, it does not -- the text is wrong. We use the convention that "streaming support" implies being able to compute the output at time $t$ using only inputs $x_i$ where $i\leq t$. Precomputing the noise can be done independently of the input for a given upper bound $T$, and therefore poses no obstacle to supporting streaming. We will correct the writing in the revised version: Henzinger et al.'s algorithm can release $T$ prefix sums in time $O(T\log T)$, but the space needed is much higher than for our method.
>
> *Q3: [...] is there any (informal) reason why one should think that it is hard to implement their algorithm using say polylogarithmic space and time?*
>
> Their algorithm’s output at time $t$ requires the computation of a linear combination of $t$ random choices. The coefficient of each past random choice changes at each new time step, making it (seemingly) impossible to (productively) reuse past linear combinations to construct new ones. Because of this, it seems likely that you are forced to store all $t$ random choices in memory to produce the output at time $t$, which bars a polylogarithmic space/time implementation.
>
> *Q4: Can you not leverage the binary representation of the leaves that store an $x_i$ to have at most $h/2$ zeros and the binary representation of every queried time step leaf to have at most $h/2$ ones?*
>
> You are correct, that would allow you to reduce the height of the tree while keeping our constant improvement in the leading term as an upper bound. We investigated this but opted to not include it as:
> 1. We wanted to make our algorithm as simple as possible.
> 2. The improvement in the leading term for the variance at a given time step is unchanged.
> 3. The reduction you get in height is not that substantial.
> 4. Although, as you correctly point out in *Q1*, we could pad with Gaussian noise to achieve identical error distribution at each step for Gaussian mechanisms, we cannot do the same for non-stable noise distributions.

---

> > ### Comment · Reviewer_aGJ1 · 2023-08-12
> >
> > Thank you the authors for the response.
> >
> > Regarding Q1 (and Q4.4), I don't actually believe that the trick requires stability of the distribution. For example, let's consider the standard binary tree mechanism. Let's say the noise added to each node of the tree is drawn from a distribution $\mathcal{D}$ (e.g. Laplace). Then, the noise for the output at time $t$ is $k_t$-fold convolution of $\\mathcal{D}$ for some non-negative integer $k_t$. Let $k_{\\max} = \\max_t k_t$. We can simply drawn $k_{\\max} - k_t$ samples from $\\mathcal{D}$ and add it to the output at time $t$. This way, the noise at every time step is identically distributed.
> >
> > Regardless of the above, I'd still strongly encourage the authors to clarify W2 and Q1 in more detail in the revision.

---

> > > ### Author Response · Authors · 2023-08-14
> > >
> > > Thank you to the reviewer for further comments.
> > >
> > > We agree with the reviewer -- our primary contribution is an improvement in the variance of the binary mechanism while retaining its computational efficiency. Our algorithm also achieves an error that is identically distributed at each step at no additional expense, but, as you correctly point out, this property can be implemented for other existing algorithms and for arbitrary distributions.
> > >
> > > We will make this point clearer in the revision, and we will also expand on W2.

---

### Official Review · Reviewer_kh8m · 2023-07-07

**Soundness:** 3 good
**Presentation:** 2 fair
**Contribution:** 4 excellent
**Rating:** 6
**Confidence:** 3

**Summary:**

This paper studies the important problem of optimizing continual observation algorithms while also making them efficient.  There has been some recent advances in the classical binary mechanism for releasing a streaming counter subject to differential privacy.  One of the main issues with the binary mechanism is that is add varying levels of noise to each count, since the number of noise terms that are added at each round depends on the number of 1’s in the binary representation of the length of the stream.  Work from Henzinger et al. shows how the variance of the noise need not change drastically at each round and overall the variance can be significantly reduced.  However, this approach has runtime that scales linearly with the length of the stream, which can be prohibitive, especially when wanting to release counts in a streaming setting.  This work shows that it is possible to get closer to the benefits of Henzinger et al. while also having runtime that is similar to the binary mechanism, making it a great replacement for the classical binary mechanism.

**Strengths:**

There are impressive results with this work, where variance is almost constant across rounds and the overall variance is similar to the optimal approaches.  Table 1 nicely summarizes the results and how they compare to existing approaches.

**Weaknesses:**

The new algorithm in Section 3.2 is not clearly presented.  As this is the main contribution of this work, I would expect the algorithm to be clearly presented, but it is pushed to the supplementary file.  There is a nice figure to explain the binary mechanism, but not the new algorithm.  Providing an example here would help, as is done with the binary mechanism.  I even went to the supplementary code file to better understand what the algorithm is doing, but there are few comments so it is not very readable.  Will increase my score if the paper can be updated with a better description of the main algorithm.


**Questions:**

Can you give a better presentation for the algorithm and how it works?  The whole algorithm seems to only be two itemized bullets in lines 214 and 215.

---

> ### Author Rebuttal · Authors · 2023-08-09
>
> We thank the reviewer for their reading and positive comments.
>
> *Q1: Can you give a better presentation for the algorithm and how it works?*
>
> We will add further details about how the algorithm works, including a graphical illustration of how the algorithm functions reminiscent of Figure 1(b) for the conventional binary mechanism, to our revised paper. See the PDF in our global response where we have added two such examples where we compute $\mathcal{M}(2)$ and $\mathcal{M}(5)$ when $T=5$.

---

> > ### Comment · Reviewer_kh8m · 2023-08-19
> >
> > Thanks for including the figure.  I have read the rebuttal and will keep my score.

---

### Official Review · Reviewer_3m6M · 2023-07-13

**Soundness:** 3 good
**Presentation:** 3 good
**Contribution:** 3 good
**Rating:** 7
**Confidence:** 5

**Summary:**

This paper studies the problem of releasing private prefix sums in the continual release model of differential privacy. A continual release mechanism for prefix sums receives a binary stream $x_1, x_2, x_3, ...$ one element at a time and continually outputs, at each time step, the number of ones received so far.  This problem can be solved by deploying the binary tree mechanism introduced by the concurrent works of Dwork et. al. and Chan et al. from 2010. In this manuscript the authors provide a variant of the binary tree mechanism that preserves its asymptotic space complexity while achieving better variance than the base mechanism. While the variance of the proposed smooth binary tree mechanism is higher than that of the matrix mechanism proposed by Henzinger et. al., the smooth binary tree scales better in time and space, with respect to the dimensionality of the stream elements. Finally, the authors claim that the error of this mechanism is identically distributed for all outputs of the mechanism.

**Strengths:**

The problem of private prefix sums is a fundamental problem which is used as a subroutine in various private algorithms. Therefore the smooth binary tree mechanism with improved variance and similar asymptotic space and time complexity is an interesting contribution. The paper provides a clear explanation of the proposed mechanism as well as the naive binary tree mechanism. These explanations are supported by helpful diagrams.

**Weaknesses:**

-The text in the paper is very clear but the formal portions of the paper are (in several instances) not well specified.

-While the proposed mechanism is nice, the analysis of the mechanism could use much more detail.

-In particular, the paper makes several claims about the error of the proposed mechanism in the text of the paper, even adding this value to a table in the introduction. However the manuscript does not specifically define their notion of error or make any formal verifiable statements about the error of the proposed mechanism in either the main paper or the supplementary material. While the notion of error they are talking about can be inferred from the text, the distribution of the error of the mechanism is key selling point that the authors mention several times. This omission and the lack of detail in the analysis would make the paper fairly inaccessible for many readers.

**Questions:**

Is there a reason that you do not consider any specific notions or provide any formal statements and proofs about the error of the smooth mechanism?

In line 221 you say that the maximum sensitivity is proportional to $k$. I understand that this is true for your choice of $k=h/2$ but as far as I can tell, it need not be true in general, right? (Since the maximum sensitivity of a particular node is technically proportional to $h-k$, which need not be proportional to $k$?)

-------------------

-------------------

Typo(s) and minor comments:

On line 112 you say “let $\mathsf{bin}(n)$ be padded with leading to $h$ digits.” This sentence is missing the word zeroes.

There seems to be a mistake in line 68: It says that the sensitivity of some node $x_t$ is the number of zeroes in  $\mathsf{bin}(t-1)$. Isn't it actually $\mathsf{bin}(t-1) + 1$? (Also, line 68 occurs prior to the definition of $\mathsf{bin}(\cdot)$ which doesn’t appear till line 113.) A similar problem to that in line 68 occurs in the statement of proposition 3.1 on line 190, which is a formal statement of line 68.

In table 1, the column ‘identically distributed’ does not indicate that it is about error, or which error notion(s) it is about.

**Limitations:**

yes

---

> ### Author Rebuttal · Authors · 2023-08-09
>
> We thank the reviewer for their reading and questions.
>
> *W1: [...] formal portions of the paper are (in several instances) not well specified.*
>
> We disagree with the claim that the formal portions are not well-specified, see answer to Q1 below.
>
> *W2: [...] analysis of the mechanism seems incomplete.*
>
> Can you elaborate on where you feel that the analysis is incomplete?
>
> *Q1: Is there a reason that you do not consider any specific notions or provide any formal statements and proofs about the error of the smooth mechanism?*
>
> We disagree that we do not consider any specific notion of error and that we lack formal statements. Our error at a given time step is Gaussian with mean zero, meaning that it is fully specified by its variance. As written in lines 133-135 and further motivated in lines 149-153, we are interested in optimizing the variance at each single time step. The main theorem expresses the properties of our mechanism in these terms too. Since our mechanism has constant variance at each time step, the expected $\ell_p$ norm can be expressed as a function of the variance.
>
> *Q2: In line 221 you say that the maximum sensitivity is proportional to $k$. I understand that this is true for your choice of $h=k/2$ but as far as I can tell, it need not be true in general, right? (Since the maximum sensitivity of a particular node is technically proportional to $h-k$, which need not be proportional to $k$?*
>
> No, our claim is true in general. $k$ is the number of 0s in $\mathrm{bin}(i)$, so therefore $k$ describes the number of ancestors to the corresponding leaf that are “left-children”. Only left-children are ever used, therefore the squared $\ell_2$-sensitivity for that leaf is equal to $k$. On the other hand, $h-k$ gives the corresponding number of 1s in $\mathrm{bin}(i)$, which fixes the number of nodes added together when the path to said leaf is used to compute a prefix sum.
>
> *Q3: There seems to be a mistake in line 68: It says that the sensitivity of some node $x_t$ is the number of zeroes in $\mathrm{bin}(t-1)$. Isn't it actually $\mathrm{bin}(t-1)+1$?*
>
> The text is correct, it is the number of 0s in $\mathrm{bin}(t-1)$. It would be $\mathrm{bin}(t-1)+1$ if the root node was ever used, but our analysis assumes it is not, as stated in lines 184-185.
>
> *Q4: Also line 68 occurs prior to the definition of $\mathrm{bin}(\cdot)$ which doesn’t appear till line 113.*
>
> We opted to give a (partial) definition on line 66 in this initial section outlining the technical ideas.

---

> > ### Comment · Reviewer_3m6M · 2023-08-18
> >
> > *Thank you for your responses! I agree that this is a nice result and that its simplicity is part of its strength -- I have updated my review accordingly.*
> >
> > >The text is correct, it is the number of 0s in bin(t-1). It would be bin(t-1)+1 if the root node was ever used, but our analysis assumes it is not, as stated in lines 184-185.
> >
> > Consider a tree with 16 leaves. Now consider node 1000: The number of zeroes in bin(0111) is one, however, changing the value of bin(1000) would change the value of two internal left nodes (not counting the root node or the leaf itself.)

---

> > > ### Author Response · Authors · 2023-08-20
> > > **Reply to technical comment**
> > >
> > > Thanks for updating your review!
> > >
> > > About your example: Note that the inputs are numbered $t=1,\dots,T$ while the binary numbers start with zero. For this reason $x_t$ is stored in the leaf corresponding to the binary representation of $t-1$. In your example, changing the value of leaf bin(1000) would correspond to changing $x_9$, not $x_8$.

---

### Author Rebuttal · Authors · 2023-08-09

In response to reviewer kh8m's request for a better explanation of how the final algorithm, Algorithm 2, works, see the attached PDF below containing two graphical examples of how the algorithm computes $\mathcal{M}(2)$ and $\mathcal{M}(5)$ when $T=5$.

We commit to including at least one of the examples in the revised paper.

---

### Decision · Program_Chairs · 2023-09-21

**Decision:**

Accept (poster)

**Comment:**

The reviewers were all positive about this paper and its contributions, and felt that the authors sufficiently addressed all their concerns in the discussion phase.